# DeLTA: Automated cell segmentation, tracking, and lineage reconstruction using deep learning

**Jean-Baptiste Lugagne**, **Haonan Lin**, **Mary J. Dunlop** *

Department of Biomedical Engineering, Boston University, Boston, Massachussets, United States of America

* mjdunlop@bu.edu

**Data Availability Statement:** All relevant data are within the manuscript and its Supporting Information files.

**Funding:** This work was supported by the National Institutes of Health grants R01AI102922 and

## Abstract

Microscopy image analysis is a major bottleneck in quantification of single-cell microscopy data, typically requiring human oversight and curation, which limit both accuracy and throughput. To address this, we developed a deep learning-based image analysis pipeline that performs segmentation, tracking, and lineage reconstruction. Our analysis focuses on time-lapse movies of *Escherichia coli* cells trapped in a "mother machine" microfluidic device, a scalable platform for long-term single-cell analysis that is widely used in the field. While deep learning has been applied to cell segmentation problems before, our approach is fundamentally innovative in that it also uses machine learning to perform cell tracking and lineage reconstruction. With this framework we are able to get high fidelity results (1% error rate), without human intervention. Further, the algorithm is fast, with complete analysis of a typical frame containing ~150 cells taking <700msec. The framework is not constrained to a particular experimental set up and has the potential to generalize to time-lapse images of other organisms or different experimental configurations. These advances open the door to a myriad of applications including real-time tracking of gene expression and high throughput analysis of strain libraries at single-cell resolution.

## Author summary

Automated microscopy experiments can generate massive data sets, allowing for detailed analysis of cell physiology and properties such as gene expression. In particular, dynamic measurements of gene expression with time-lapse microscopy have proved invaluable for understanding how gene regulatory networks operate. However, image processing remains a key bottleneck in the analysis pipeline, typically requiring human intervention and *a posteriori* processing. Recently, machine learning-based approaches have ushered in a new era of rapid, autonomous image analysis. In this work, we use and repurpose the U-Net deep learning algorithm to develop an image processing pipeline that can not only accurately identify the location of cells in an image, but also track them over time as they grow and divide. As an application, we focus on multi-hour time-lapse movies of bacteria growing in a microfluidic device. Our algorithm is accurate and fast, with error rates near 1% and requiring less than a second to analyze a typical movie frame. This increase in

R21AI137843 and by the Office of Science (BER) at the U.S. Department of Energy grant DE-SC0019387. The funders had no role in study design, data collection and analysis, decision to publish, or preparation of the manuscript.

**Competing interests:** The authors have declared that no competing interests exist.

speed and fidelity has the potential to open new experimental avenues, e.g. where images are analyzed on-the-fly so that experimental conditions can be updated in real time.

## Introduction

Time-lapse microscopy is an essential technique for studying dynamic cellular processes. With automated microscope hardware and microfluidic devices it is possible to parallelize experiments to both increase data resolution and to test many different conditions in parallel. Technical improvements in hardware as well as open microscopy software initiatives [1,2] have also made time-lapse acquisitions both faster and more flexible. Researchers today can test up to hundreds of conditions [3] and strains [4] in a matter of hours and, after analysis, iteratively refine their hypotheses and design a new suite of experiments. As an initial test case, we focus our analysis on images from the so-called "mother machine" microfluidic device [5]. In this device thousands of single bacterial cells are trapped independently in one-ended growth chambers where they can be observed for extended periods of time, while their progeny are progressively flushed out of the field of view (Fig 1A). This experimental set up has been widely adopted as a popular standard for long-term, single-cell time-lapse imaging of bacteria such as *E. coli* [6–9], *Bacillus subtilis* [7,10], and *Corynebacterium glutamicum* [11]. Unfortunately, analysis of raw single-cell microscopy images remains a major bottleneck despite major efforts in this area.

While a plethora of software suites have been developed for single-cell segmentation and tracking [12–15], including code specific to analysis of mother machine data [11,16–19], the vast majority require manual inputs from the experimenter and are designed for *a posteriori* processing. The relatively recent breakthrough in biomedical image analysis brought by deep convolutional neural networks, and the U-Net [20] architecture in particular, has introduced an era of fast-paced developments in the field [21]. Deep learning-based image processing is fast, as it can be run on graphical processors. Further, it can adapt to new data after being trained, thus improving performance and robustness. In addition, as long as a reasonably large and accurate training set can be generated, the same code can be re-used without parameter or code tweaking for different experimental setups or even different organisms.

But cell segmentation alone does not extract the rich dynamic information contained in single-cell resolution time-lapse movies. A recent study demonstrated the possibility of using deep learning methods to track single, non-dividing cells over time after segmenting them [22]. However typical time-lapse movies tend to feature several division events per frame, and common object-tracking deep learning solutions cannot be used to reconstruct those lineages as they are not meant to identify divisions of the monitored objects. The possibility to robustly segment cells, track them, and reconstruct lineages on-the-fly would not only speed up analysis and make it possible to process large amounts of data, but also would enable the development of "smart" microscopy platforms that could automatically trigger actions such as microfluidic or optogenetic inputs based on cellular events like divisions or transcription bursts.

As an example, a small number of recent studies have highlighted the potential of computer-based feedback control of gene expression in single cells as a new experimental paradigm [9,23,24]. This approach automatically adjusts chemical or optogenetic inputs based on real-time quantification of gene expression levels in cells to precisely and dynamically perturb regulatory networks. These studies have garnered insights into the dynamics of cellular processes that would have been impossible to study by other means [25]. However, to be able to perform feedback control at the single-cell level, image analysis must be performed on-the-fly, without

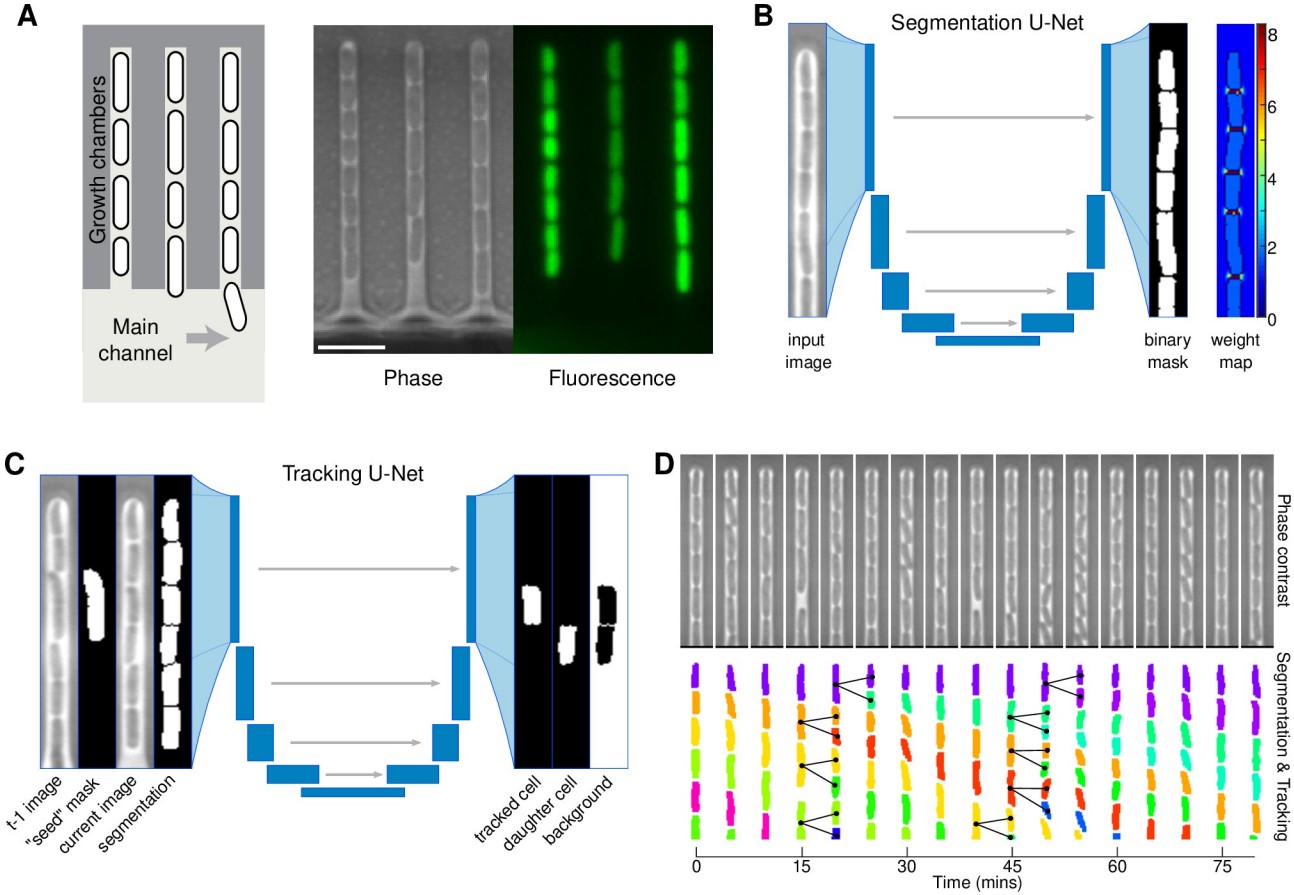

**Fig 1. Core elements of the DeLTA pipeline and segmentation and tracking results.** (A) Schematic representation of mother machine device. Mother cells are trapped at one end of the chamber and their progeny are progressively flushed out of the chamber, into the main nutrient flow channel. Fluorescent reporters can be used to monitor single-cell gene expression. Scale bar is 5μm in length. (B) Inputs and outputs of the segmentation U-Net. Note that the weight maps are only used for training and are not produced at the prediction step. (C) Inputs and outputs for the tracking U-Net. (D) Representative kymograph of segmentation and tracking for a single chamber. Black lines highlight detected divisions and mother-daughter relationships.

any human intervention. To circumvent this problem, researchers have exploited constrained experimental geometries to localize cells [9], developed customized interfaces with flow cytometry [26,27], or restricted studies to experimental durations that avoid too many cell division events or conditions where the cells escape from the field of view [23]. By providing rapid access to a suite of quantitative, dynamic measurements about single cells, entirely automated image processing offers the potential to expand the scope of experiments that can exploit dynamic, real-time cell tracking.

Here we introduce DeLTA (Deep Learning for Time-lapse Analysis), an image processing tool that uses two U-Net deep learning models consecutively to first segment cells in microscopy images, and then to perform tracking and lineage reconstruction. After training, our pipeline can analyze new acquisitions in a completely automated fashion, with error rates of 0.06% for segmentation and 1.01% for tracking and lineage reconstruction. Data analysis is fast, requiring between 300-800msec per frame on consumer-grade hardware, depending on the number of tracked cells (Table 1). We used a diverse set of training samples, both from our own experiments and from data retrieved from the literature, in an effort to generate models

**Table 1. Key performance numbers for training and evaluation of the DeLTA algorithm.**

| Training | | |
|---|---|---|
| | Segmentation | Tracking |
| Set size | 8,258 chambers (~53,000 cells) | 7,706 tracking events |
| Set construction time | ~10 hours | ~10 hours |
| U-Net training time | 3 hours 20 minutes | 8 hours |
| Evaluation | | |
| | Segmentation | Tracking |
| Ground-truth set size | 3,422 cells | 3,073 cells |
| Errors (rate) | 2 (0.06%) | 31 (1.01%) |
| Sample processing time | 12 msec (/chamber) | 3.6 msec (/cell) |
| Frame processing time | 212 msec | ~400 msec (depending on # of cells) |
| Movie processing time (262,634 cells) | 9 minutes 2 seconds | 13 minutes 8 seconds |

for DeLTA that can perform out-of-the-box without further training on entirely new mother machine data. In addition to the deep learning algorithm itself, we also introduce a suite of scripts and graphical user interfaces to interface our Python-based U-Net implementation with Matlab to create and curate training sets. The code is available on Gitlab to facilitate distribution and adaptation by other researchers (Methods). By capitalizing on recent advances in deep learning-based approaches to image processing, DeLTA offers the potential to dramatically improve image processing throughput and to unlock new automated, real-time approaches to experimental design.

## Results

We applied the code directly to the problem of segmenting *E. coli* cells trapped in mother machine microfluidic chambers [5]. In the device, cells are grown in chambers of ~1μm in width and height, and 25μm in length (S1 Fig). This configuration traps a so-called "mother" cell at the dead-end of the chamber, and as the cells grow and divide the daughters are pushed out of the other end of the chamber (Fig 1A). Nutrients are introduced by flowing growth medium through the main channel, which also serves to flush out daughter cells. This device has become a popular way to study single-cell dynamics in bacteria, as it allows for the long-term observation of single mother cells, with typical experiment durations of 12 to 30 hours [4,6,7,9], and sometimes up to several days [10]. It is also possible to image progeny, allowing for comparison of daughters, granddaughters, and great granddaughters for multi-generation analysis. This latter source of data is less frequently exploited due to the challenges associated with accurate tracking, therefore analysis has traditionally focused on the mother cell. However, these data have great potential, as they can contain generational information for thousands of cells. The DeLTA algorithm we introduce here is centered around two U-Net models, which are trained on curated data and then can be used to quickly and robustly segment and then track cells in subsequent images.

### Segmentation

The first U-Net model (Fig 1B, S2 Fig) is dedicated to segmentation and is very similar to the original model [20]. To create training sets, we began by using the Ilastik software [14] to segment time-lapse movies of cells in the mother machine chips. Note that other training set generation pipelines can be used, for example with other mother machine data analysis software [11,16,17,19], or if experimenters have pre-existing segmented sets. While our segmentation

results with Ilastik were already fairly accurate (~1% error rate), we found that even after carefully designing an initial training set, the results generalized poorly to new datasets and usually required training sets to be generated or updated *a posteriori* with each new experiment. Therefore, we used a combination of Ilastik and manual curation to produce segmentation outputs that then served as training sets for our code. We stress that this training set generation process does not need to be repeated after initial training.

Our code uses data augmentation operations (Methods) to ensure robustness to experimental variation and differences in imaging conditions. For example, changes in nutrient concentration can produce cells of different size or aspect ratio, or subtle light condenser misalignments can introduce differences in illumination of the image. With simple image transformations or intensity histogram manipulations to approximate such changes, the model can be trained to generalize to new, different inputs.

To generate the training sets, the time-lapse movies were cropped into thousands of pairs of images and segmentation masks for each chamber within the mother machine at each time point (Methods). These potential training samples were then manually curated with a rudimentary graphical user interface in Matlab to ensure that the U-Net algorithm was only fed accurate training samples (S3 Fig). We curated a sufficient training set (~8,300 samples), which took approximately two days to generate, and then trained the Python-based U-Net model against the segmented binary masks (Fig 1B).

To train the network we implemented a pixel-wise weighted binary cross-entropy loss function as described in the original U-Net paper [20] to enforce border detection between cells. Prior to data augmentation and training, weight maps were generated to increase the cost of erroneously connecting two cells together. In addition to the original procedure to generate the weight maps described in the U-Net paper, we also introduced an extra step that sets the weight to zero around the 1-pixel contour of the cells. By doing so, we relax the constraint on learning of the exact contour as provided in the segmentation masks in the training set. This makes the model more generalizable, as the algorithm does not have to exactly fit the output of the Ilastik segmentation.

The network was trained on images from three different multi-position time-lapse movies acquired in our lab, and on movies available from the literature [11,16,17,28,29]; we then evaluated on a third data set acquired in our lab weeks later as well as on data from the literature (Table 1). To demonstrate generalizability of the approach, and to evaluate performance against other analysis software, we also used the time-lapse image datasets provided with the mother machine analysis software tools Molyso [11], MoMA [16], and BACMMAN [17] to evaluate the performance of DeLTA on completely new data by performing "leave-one-out" training and evaluation pairings. In these studies, one of those three different datasets was left out at the training step, and after training, performance was evaluated against the omitted dataset (S1 Table).

DeLTA performed extremely well when segmenting bacterial images, with only 2 errors out of 3,422 (0.06%) segmented cells when compared to our ground-truth evaluation set (Table 1, S4 Fig). With our desktop computer, each frame in our evaluation movie was processed in 212ms. The evaluation movie that we acquired ourselves, which features 12 positions, each with 193 frames was processed in about 9 minutes. It also performed remarkably well on the BACMMAN and Molyso datasets with an error rate of 0.22% (4 errors out of 1785 cells) and 0.21% (4 errors out of 1874 cells), respectively (S1 Table). Performance on the MoMA movie, which looks markedly different from the other movies in our datasets was poor, with a segmentation error rate of 17.3% (179 errors for 1037 cells). We emphasize that for these results, the U-Net models had not been trained on samples from the BACMMAN, Molyso, or MoMA datasets before processing them. Therefore, our results with the BACMMAN and Molyso

datasets are excellent; with incorporation of training sets that look similar to the MoMA data we anticipate that performance on those movies would dramatically improve.

## Tracking and lineage reconstruction

The main innovation in our approach is to use a U-Net architecture not only to segment cells, but also to track them from one frame to the next and identify cell divisions. To our knowledge, this is the first time a deep learning model has been used to accomplish this second half of the time-lapse movie analysis pipeline. Using an approach that mirrors the original U-Net architecture (Fig 1C, Methods), we use multiple images as inputs and outputs. Namely, for every cell at every time-point, we use four images as inputs: the transmitted light images for the current frame and the previous frame, a binary mask delimiting one "seed" cell that we want to track from the previous frame, and the segmentation mask for the current frame. This second U-Net model is used downstream of the segmentation U-Net to complete our time-lapse analysis pipeline. As outputs for training, we provide two binary masks, the first one delimiting where the seed cell is in the new frame, and the second delimiting where the potential daughter cell is, in case a division has happened (Fig 1C).

We developed a simple graphical user interface in Matlab for creating training sets. Because manually creating a dataset for this tracking step is not as tedious as for segmentation, we did not incorporate third party software into our workflow. Instead, the user simply clicks on cells to identify them in the new frame. Using this method, we generated a sufficient dataset with ~7,700 samples in about two days. Afterwards, in order to perform the same type of leave-one-out analysis we used to evaluate the segmentation step, we used ground-truth movies we generated for the BACMMAN, Molyso, and MoMA datasets as training samples for tracking, and compiled training sets containing all but one of the datasets. Other approaches could speed up this process, such as employing tools already available for automated tracking [13,14,16,30]. However, users must be particularly careful not to introduce erroneous training samples into their set. To this end we also provide a graphical user interface similar to the one for segmentation that can be used to curate training samples. Note that it can also be used to curate and re-purpose the prediction output of the tracking U-Net as a training set, allowing for an iterative approach to training set generation. Again, we found that training does not need to be repeated provided the general image analysis pipeline and cell morphology remain the same. Once the tracking U-Net has generated tracking predictions on new data, the binary mask outputs are compiled into a more user-friendly lineage tree data structure. We provide a simple Matlab script that loops through the tracking mask outputs of the deep learning pipeline to assemble this structure.

Once trained, we applied our tracking pipeline to follow cells and identify cell divisions in evaluation time-lapse movies that it had never seen before (Fig 1D). Tracking and lineage identification errors rates were low, with only 31 mistakes in an evaluation set of 3,073 samples (1.01%). The majority of those errors arose from either non-trivial tracking problems where humans also had difficulties identifying cells from one frame to the next or were associated with cells at the bottom of the image that were in the process of being pushed out of the field of view (S5 Fig). Processing times are fast, with each cell in our evaluation movie tracked in 3.6ms and the 262,634 cells identified at the segmentation step processed in about 13 minutes. Following leave-one-out tracking performance analysis, we were surprised to see that tracking error rate for the BACMMAN dataset was even lower than for the movies generated in our lab, with 0.53% (8 errors out of 1504 cells). Although the BACMMAN movie appears less complicated than our own evaluation movie, it is notable that we were able to obtain high fidelity results on data DeLTA was not trained on. Results for the Molyso dataset were also acceptable, with a 3.03% tracking error rate (46 errors for 1514 cells). Tracking on the MoMA dataset was

1.85% (4 errors for 216 cells), though we note that this result is not as relevant since so many errors were introduced in the segmentation stage. Again, these results are likely to improve if the tracking model is trained on similar images.

## Feature extraction

In addition to the segmentation and tracking models, we also developed a Matlab script for extracting morphological features such as cell length, area, and fluorescence levels following segmentation, tracking, and lineage reconstruction. As an example, we show results from a single chamber within the mother machine where we track a mother cell and its progeny over the course of 16 hours (Fig 2A). The *E. coli* in this movie contain a green fluorescent protein (GFP) reporter (Methods). For each cell, we extracted mean GFP levels over time and also obtained morphological data. The length of the mother cell over time shows characteristic growth and division patterns (Fig 2B). These data were also used to calculate growth rate (Fig 2C) and the timing of cell division events (Fig 2D). Division events allow us to track daughters, grand-daughters, and great granddaughters before they are flushed out of the chamber (Fig 2E).

To demonstrate the wealth of information that can be extracted from time-lapse movies using our analysis pipeline, we compared the fluorescence levels between different generations. We compared the mean GFP of the mother in the time interval between the preceding cell division and the current cell division to the mean GFP of the daughter in the interval between the current division and the next division (Fig 3A). As expected, there was a strong correlation between mother and daughter fluorescence levels. We extended this analysis to granddaughters or great granddaughters by comparing GFP levels before the n-1 or n-2 division event for the grandmother or great grandmother (i.e., the last period of time the two cells were the same cell) to GFP levels after the current (n) division. We observed a decrease in the correlation between fluorescence levels the further away cells were in generations (Fig 3B). Highlighting the potential for massive data collection, the analysis for this specific fluorescent reporter data

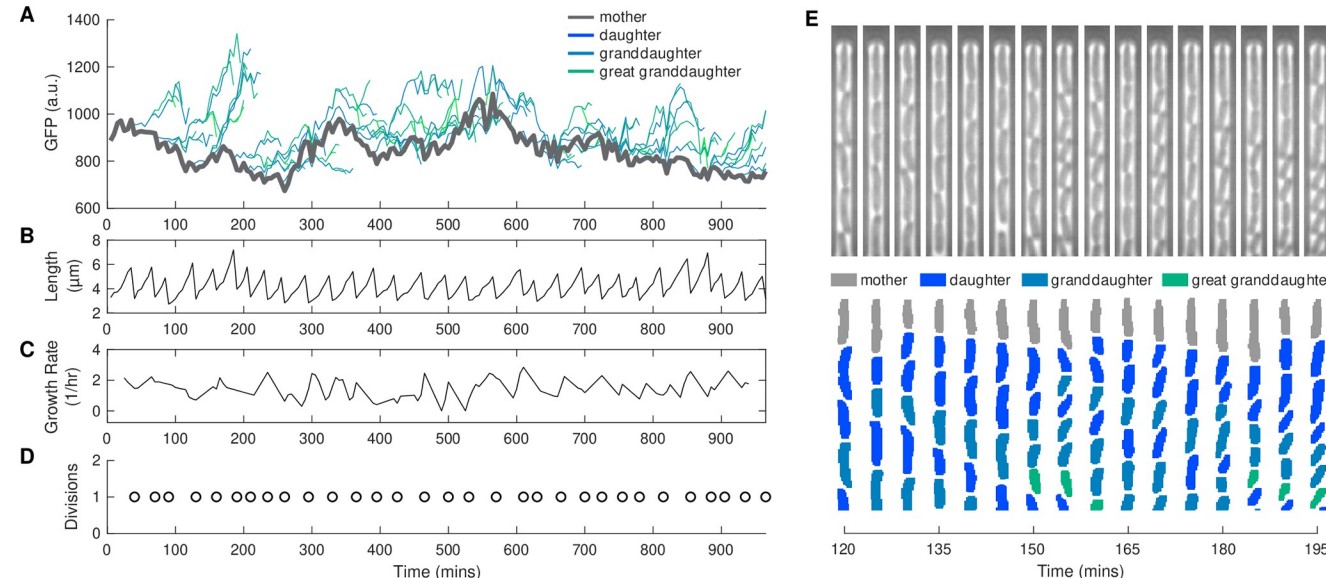

**Fig 2. Representative single-cell time-lapse image data demonstrating wealth of information extracted from movies.** (A) Mean GFP fluorescence over time for mother cell and its progeny. (B) Cell length, (C) growth, and (D) timing of cell division events over time for the mother cell. Note that these morphological properties are also recorded for all progeny, but are not displayed here for visual clarity. (E) Kymograph of chamber containing mother cell and progeny presented in (A – D).

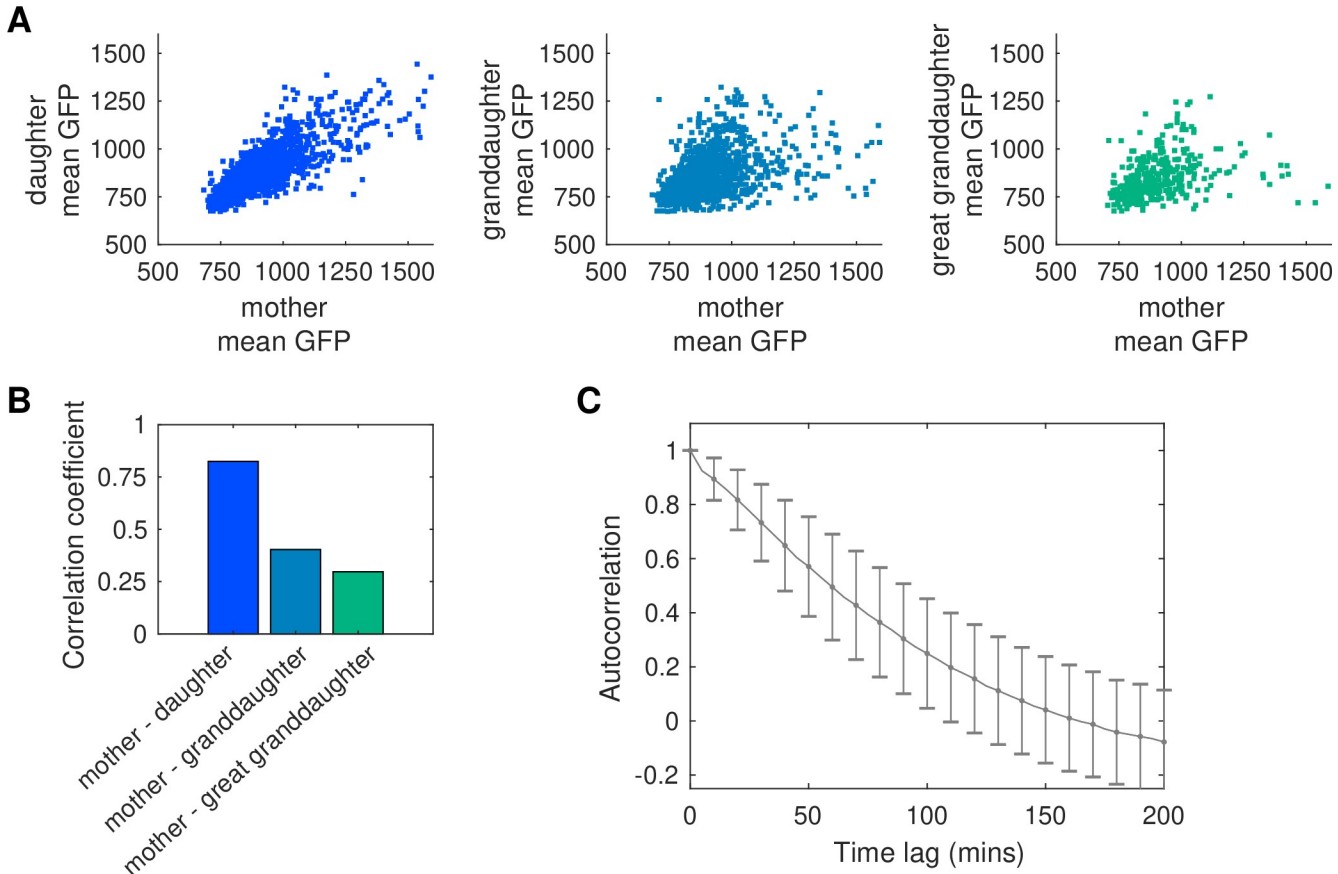

**Fig 3. Analysis of fluorescence correlations in the lineage tree.** (A) Mean GFP fluorescence for the mother cell compared to daughter, granddaughter, or great granddaughter cells. Fluorescence values for the mother are derived from the cell cycle immediately prior to division, while fluorescence for the daughter, granddaughter, or great granddaughter come from the cell cycle immediately following division from the cell in the previous generation (e.g. from the mother when considering the daughter). (B) Correlation coefficient between mother cell mean GFP values and its progeny. (C) Autocorrelation of the GFP signal for the mother cell. Error bars show standard deviation around the mean.

includes 10,793 mother-daughter comparisons, 7,932 mother-granddaughter comparisons, and 2,078 mother-great granddaughter comparisons from a movie that ran for 16 hours with 108 different chambers.

In addition, imaging many mother cells over long durations provides high quality temporal data. We illustrate this by calculating the autocorrelation of the GFP signal (Fig 3C). The resulting data give a high-fidelity view into the correlation times associated with the reporter, underscoring the potential for highly accurate measurements of dynamic, single-cell properties. Indeed, a rapidly decreasing autocorrelation would indicate that the signal GFP reports fluctuates rapidly over time, and that much of the variability in the population emerges from intrinsic noise at the single-cell level. In turn this can give us insights into, for example, diversification and bet hedging strategies, which could not be obtained without reliable, long-term single-cell segmentation and tracking [28,31].

## Discussion

In this study, we present an image analysis pipeline centered around two deep learning models based on the U-Net architecture to segment, track, and identify divisions of bacteria growing in

a microfluidic device. The U-Net architecture has proven to be a breakthrough technology for cell segmentation and is emerging as a standard for this image analysis task. With our training set and implementation, the error rate for segmentation drops to an impressive 0.06%. In addition, our novel approach extends these ideas to apply the deep learning model to also track cells and identify division events. The 1.01% error rate on this task is excellent and a clear improvement over other software designed to segment and track bacteria in mother machine devices. For instance, the mother machine movie analysis software MoMa [16] reports tracking error rates of about 4%. Other proposed solutions do not explicitly report their accuracy and typically require some parameter tweaking to be adapted to new data, and often also require *a posteriori* correction [10,11,17,19,32]. In contrast, DeLTA can adapt to entirely new images from a different setup without the need for re-training if those new images are similar to those in the training set. We note that our algorithm for post-processing the tracking output of our deep learning pipeline is very simple, and more elegant rule-based and optimization algorithms proposed in other studies [16,17,33] could be integrated with our approach to reduce the error rate even further. Additionally, we provide large training datasets that can be used by others to improve upon our work, and we have incorporated data from other groups in our training sets to make our trained models as generalizable as possible. As a future direction, it would be helpful to explore stopping criteria to prevent overfitting by using diverse and different training and validation sets.

Indeed, we anticipate that the deep learning workflow presented here can be applied to any other microscope and mother machine devices without any code or parameter modifications. Other researchers can generate their own training sets following the workflow described in Results and Methods. The tedious construction of training and evaluation datasets is often a limiting step in the adoption of machine learning techniques. To this end we provide simple scripts and graphical user interfaces for users to assemble their own training sets for their specific setup. Beyond bacteria in mother machine-type devices, our approach can, in principle, be applied to a wide range of similar problems. We rapidly explored this possibility by applying our software to tracking of the yeast species *Saccharomyces cerevisiae* freely growing in two dimensions, with benchmarking data taken from another study [34]. Although encouraging, the tracking error rate of our pipeline was in the 5–10% range for datasets it was not trained on (S6 Fig), and thus does not outcompete state-of-the-art software designed specifically for yeast data analysis [34,35]. However, the yeast datasets we used are much smaller than what we generated for training our network on the mother machine data, which likely limits the performance of our models. Better post-processing algorithms could also help reach the same accuracy we achieved with the mother machine experiments. The ability of the algorithm to segment and track cells with dramatically different morphology and experimental constraints from the mother machine data suggests excellent potential for generalization of the approach.

In conclusion, the DeLTA algorithm presented here is fast and robust, and opens the door to real-time analysis of single cell microscopy data. We anticipate that there are a number of avenues for improvement that could push the performance even further, improving accuracy and processing speed. For instance, a clear candidate would be to use the U-Net tracking outputs as probability inputs for linear programming algorithms to compile lineages [33], replacing the simplistic approach we use in post-processing. Another possible extension to the work would be to implement a single U-Net model that performs segmentation and tracking simultaneously. In principle, this could increase processing speed even further by streamlining segmentation and tracking, and improve accuracy as the segmentation step would also take into account information from previous frames. Although we have focused on *E. coli* bacteria in the mother machine here, preliminary tests suggest that this algorithm may be broadly applicable to a variety of single-cell time-lapse movie data. We expect this approach to be generally

well-suited to high-throughput, automated data analysis. In addition, this algorithm can be incorporated in "smart" microscopy environments, which represent a promising emerging research field, as cells and computers are interfaced to probe complex cellular dynamics or to steer cellular processes [25].

## Methods

### Data acquisition and datasets

The *E. coli* strains imaged in the mother machine are an *E. coli* BW25113 strain harboring a low-copy plasmid where the native promoter for the *rob* gene drives expression of a gene for green fluorescent protein (*gfp*). The reporter comes from Ref. [36]. Cells were grown in M9 minimal medium with 0.4g/L glucose, 0.4g/L casamino acids, and 50μg/mL kanamycin for plasmid maintenance. The growth medium was also supplemented with 2g/L F-127 Pluronic to prevent cell adhesion outside of the growth chambers.

The mother machine microfluidic master mold was designed using standard photolithography techniques [37]. Polydimethylsiloxane (PDMS) was poured onto the wafer, cured overnight, and plasma bonded to a glass slide to form the final microfluidic chip. The chip features 8 independent main channels where media flows, and each channel features 3,000 chambers of 25μm in length and 1.3 to 1.8μm in width (S1 Fig, and https://gitlab.com/dunloplab/mother_machine for GDSII file and details). Three time-lapse movies, two for training and one for evaluation, were acquired with a 100X oil objective on a Nikon Ti-E microscope. A fourth time-lapse movie used for training was acquired on another Nikon Ti-2 microscope, also with a 100X oil objective. The temperature of the microscope chamber was held at 37˚C for the duration of the experiment. Images were taken every 5 minutes for 16 to 20 hours. An automated XY stage allowed us to acquire multiple positions on the chip for each experiment. The three training movies and the evaluation movie for mother machine data contain 18, 15, 14, and 12 positions, respectively, with each imaging position containing 18 chambers. For each position and timepoint, the images were acquired both in phase contrast and epifluorescence illumination with a GFP filter.

In addition to our own movies, we also analyzed movies that were provided as supplementary data sets associated with other studies in the literature. Specifically, we used datasets associated with Molyso [11], MoMA [16] and BACMMAN [17] mother machine analysis software tools as well as other mother machine movies available from the literature [28,29] to generate diverse training sets. Unless otherwise specified, such as in the case with the leave-one-out analysis, all datasets were used when compiling training sets.

All raw data and datasets are available online at:https://drive.google.com/drive/folders/1nTRVo0rPP9CR9F6WUunVXSXrLNMT_zCP

### Implementation and computer hardware

We recommend using the Anaconda Python distribution as this greatly simplifies installation and increases portability. All the necessary libraries can be installed on Windows, Linux, or MacOSX with Anaconda in just a few command lines, detailed in the Gitlab repository: https://gitlab.com/dunloplab/delta.

The U-Net implementation we use here is based on Tensorflow 2.0/Keras and is written in Python 3.6. The general architecture is the same as the one described in the original U-Net paper, with small variations in the tracking U-Net architecture to account for the different input/output dimensions and the different loss functions (S2 Fig). Our code can be run on a CPU or GPU, though GPU results will be faster. Pre-processing and post-processing scripts as well as training set curation scripts and Graphical User Interfaces (GUIs) were written for and

executed on Matlab R2018b. All computations were performed on an HP Z-840 workstation with an NVidia Quadro P4000 graphics card.

## Movie pre-processing

To reformat the movies for subsequent analysis with our U-Net pipeline, the movies were first pre-processed in Matlab. This analysis allowed us to crop the larger image into small images featuring a single growth chamber that were individually analyzed. A model image of a single empty chamber was cross-correlated with the full-view frame to detect the position of the chambers in the first frame of each movie. After cross-correlation, peaks in the cross-correlation product image were detected as the centers of each individual chamber and an image of each of them was cropped around this point and saved to disk. Afterwards, we applied a second cross-correlation operation to correct for small XY drifting errors between frames. The individual chambers were then cropped out of each frame and saved to disk for subsequent analysis.

## Training set generation

The time-lapse movies used for training were first analyzed with the Ilastik software [14] to produce potential segmentation training samples (S3 Fig). Image pixels were categorized into three classes: Cell insides, cell contours, and background (anything that was not a cell pixel). By using the two different classes for the inner part of the cells and the contour, we minimized under-segmentation errors where two touching cells would be connected in the segmentation binary mask. The Ilastik output was then processed with a custom Matlab script that applied simple mathematical morphology operations to get rid of small erroneous regions, and then used watershedding to expand the inner part of each cell into the border pixel regions. The resulting binary segmentation masks cover the entire cell area but do not tend to connect independent cell regions together. The chamber cropping and XY drift correction operations described above were applied to the Ilastik output. Ilastik project files are provided with the rest of the data as examples.

A rudimentary GUI was then used to manually curate a random subset of the Ilastik segmentation samples to generate a training set. Once a large enough training set was generated (we started obtaining reliable results with around 1,000 chamber segmentation samples for the *E. coli* mother machine data, however this number will vary with the complexity of the task), pixel-wise weight maps were generated (Fig 1B). Different parts of each training mask were weighted based on the formula described in the original U-Net paper [20], where a strong emphasis is put on "border pixels", i.e. pixels that lie on a thin border region separating two cells. Combined with the pixel-wise weighted cross entropy described below, this extra weight on border pixels forces U-Net to learn those separations and not under-segment two cells that are touching. We then added an extra step where we set the weight for contour pixels, i.e. pixels on the outer perimeter of cells in the binary mask, to 0. The exact contour of cells is hard to determine, even for humans, and we found that the Ilastik approach can introduce artifacts in the exact cell contour. To prevent overfitting U-Net to exactly the cell contours produced by our training set generation method, we added this 1-pixel margin. Using this step marginally improves performance, with an error rate on our evaluation movie of 0.09% without the zero-weight pixel margin used for training instead of the 0.06% error rate we obtain with it. However, because the total number of errors is very low, it is hard to conclude whether this effect is significant. It does however reduce pixel error between the ground-truth (described below) and the DeLTA segmentation output from 4.4% to 1.64%.

Once the segmentation U-Net was trained, we moved on to creating a training set for the tracking U-Net model (S3 Fig). We used the trained segmentation U-Net to predict segmentation masks on the training movies. This segmentation output was then used to generate the training set for tracking with another simple GUI. In the GUI, a random segmented cell from the training movies is presented and the user manually generates the training set by clicking on the cell in the frame at the next timepoint, as well as on the daughter cell if a division happened between those timepoints. For tracking cells in the mother machine, we generated a training set of 7,706 samples.

## Data augmentation

An important step when using deep neural networks like U-Net that contain millions of parameters to fit is to artificially increase training set size by applying random transformations to the inputs and outputs. This "data augmentation" step not only increases training set size but ensures that the model does not overfit the data and that it can easily generalize to new inputs, for example those generated with relatively different imaging conditions.

Our training data was augmented on-the-fly with custom Python generators. We implemented our own data augmentation functions as we sometimes required finer control over image manipulation operations than what the standard Keras functions offered. We used mostly standard operations such as random shifting, scaling, rotation and flipping, but we also added three non-standard operations, one for elastic deformations as described in the original U-Net paper [20] and two for manipulating image contrast to simulate variations in illumination between experiments (S7 Fig). The first image intensity manipulation operation applies a randomly-generated and monotonically increasing Piecewise Cubic Hermite Interpolating Polynomial (PCHIP) mapping operation on pixel intensities over the entire image, which in effect distorts the image histogram to simulate the effect of different lamp settings on illumination. The second image intensity manipulation operation also relies on a randomly generated PCHIP. The generated curve is multiplied along the y-axis (i.e. along the chamber axis) of a single chamber image, meant to simulate the illumination artefacts that can appear at the border between different regions of a microfluidic chip under phase contrast illumination. Without these last two data augmentation operations, the segmentation error rate against our evaluation movie ground-truth nearly quadruples from 0.06% to 0.23%.

## U-Net architecture and training

We used a standard U-Net architecture for the segmentation model (Fig 1B & S2 Fig), with 5 contraction/up-sampling levels. For training with the weight maps described above, we implemented our own loss function, as pixel-wise weighted binary cross-entropy is not a standard loss function available through the Keras API. The model was trained over 400 epochs of 250 steps, with a batch size of 10 training samples. As described above, data augmentation operations were randomly applied on-the-fly to reduce the risk of over-fitting.

The tracking model (Fig 1C & S2 Fig) is similar to the segmentation U-Net, but differs in two significant ways. First, the input layer contains four components per training sample (image of the previous frame, binary mask of the "seed" cell in previous frame, image of current frame, and binary mask of all segmented cells in current frame (Fig 1C)). Second, the output layer contains three components per training sample, with the first one corresponding to the mask of the "seed" cell in the new current frame, one corresponding to the mask of the daughter cell if a division just happened, and finally the third layer is the mask of everything else in the image. Because those three components are mutually exclusive, i.e. a "mother" pixel cannot also be a "daughter" pixel, we were able to use the standard categorical cross-entropy

loss function provided by Keras. This categorical cross-entropy determines whether each pixel is categorized as part of the tracked cell in the image, its potential daughter, or the background, and attributes misclassification costs accordingly. For each training set, the model was trained over 400 epochs of 250 steps, with a batch size of five training samples. The same data augmentation operations as described above were applied on-the-fly as the network was trained.

For both segmentation and tracking, the standard Keras Adam optimizer was used with a learning rate of $10^{-4}$.

### Prediction, post-processing and evaluation

After the tracking step, the data contained in the output images was reformatted into a more user-friendly structure using Matlab. The pixels attributed to each cell in the tracking output were matched with the pixels in the segmentation mask for the current frame, and a score matrix for this specific chamber and timepoint was generated. Conflicts, for example where two cells get the same attribution score from the same cell in the previous frame, are discarded with simple rules where one cell simply becomes a "new" cell that appears and forms a new lineage tree. The low error rate at the U-Net tracking step allows us to use such simplistic methods and still get good results, but more elegant tracking algorithms [33] using our tracking prediction maps as inputs could further increase performance. The code also extracts morphological features like single cell length and area, as well as fluorescence levels associated with the fluorescence images in the movie.

We manually created ground-truth outputs for segmentation and tracking by using the first 30 frames of the first position of our evaluation movie, the Molyso movie, and the BACMMAN movie, and for all of the 10 frames of the MoMA movie. To evaluate the performance of DeLTA on those datasets, we compiled four leave-one-out training sets, in which one of the datasets was removed entirely from the samples used to train the segmentation and tracking U-Nets. We then evaluated segmentation and tracking error rates by comparing the ground-truth from the left-out dataset to the DeLTA output on that same dataset (Table 1, S1 Table).

The first metric, which we refer to as "segmentation error rate" is the percentage of under-segmented, over-segmented, false positive, and false negative cells. To compute this, we first perform an attribution step in which cells in the DeLTA output to be evaluated are matched to cells in the ground-truth if they overlap with the ground-truth cell over at least 90% of their surface. If two or more cells are attributed to one cell in the ground-truth, this is considered as a possible over-segmentation error. However, because the exact time that two cells can be considered to have divided is arbitrary to some degree, we do not count it as an actual error if the two cells in the evaluated output just divided and the cell in the ground-truth divides in the next frame. Similarly, we do not consider one cell that is attributed to two in the ground-truth to be an under-segmentation error if the cell is about to divide and the ground-truth cells just divided. If a cell is present in the ground-truth but not in the evaluated output, we count it as a false negative, and if a cell is present in the output but not in the ground-truth, we count it as a false positive. Unless those cells are in the bottom 5 pixel margin of the image and are about to be flushed out, either in the ground-truth or the output being evaluated, we do not count these as errors, as identifying these as valid cells is also arbitrary.

To evaluate tracking performance, we simply count each time a cell is attributed to the wrong cell in the next frame as an error, and report the tracking error rate as the number of errors divided by the number of cells tracked. Tracking events that are tied to cells that were either erroneously segmented or that were analyzed as divided at an earlier or later timepoint than the ground-truth were excluded from this analysis. In most cases, this effect is minimal because segmentation errors are so rare.

### Data analysis with generation information

Fluorescence data extracted from the cell lineages were used to measure correlation coefficients (corrcoef function in Matlab) between mother cell mean GFP levels and daughter, granddaughter, or great granddaughter mean GFP levels. In this analysis, we measured the GFP level for the mother cell using the time interval immediately prior to division and considered one cell cycle worth of data. We compared this to one cell cycle worth of GFP data for the daughter (or granddaughter, great granddaughter) immediately following division from the mother cell.

Autocorrelation data (computed using the xcov function in Matlab which computes the cross-variance between two signals, or in our case of a cell's GFP signal with itself) for each mother cell were normalized to the value at zero time lag.

## Supporting information

**S1 Fig. Mother machine chip design used in this study.** This image is adapted from the mask design file of the mother machine microfluidic chip that was used to generate the training and evaluation data. The upper half of the figure shows the entire chip which features 8 independent main channels of 400μm in width and ~60μm in height in which media containing nutrients and selection markers is flown. Each of those channels features on its side 6,000 chambers of 1.1μm in height, 25 or 35μm in length, and 1.3 to 1.8μm in width in which mother cells are trapped and can then be grown for hours to days.
(TIF)

**S2 Fig. U-Net architecture for segmentation/tracking.** The layers and tensor dimensions used in our U-Net implementation. The differences between the segmentation and the tracking U-Net are highlighted with red rectangular boxes. The first element is for the segmentation U-Net version, the second for the tracking version (segmentation/tracking). Note that the architecture is the same as the original U-Net model, except for image dimensions, number of input and output layers, and the final activation layer for tracking. The values we used for these are noted on the figure. The loss function for segmentation is a pixel-wise weighted binary cross-entropy as described in the original U-Net paper. The pixel-wise weight maps are provided for each output mask during training as an auxiliary input. The loss function for tracking is the standard categorical cross-entropy function from Keras.
(TIF)

**S3 Fig. Training set construction.** (A) We used the Ilastik software to generate initial segmentation masks for training. The Ilastik software uses a random forest classifier and various local pixel features to classify pixels in an image on the basis of a training set drawn by the user. On the left side of the image are the three classes we use to generate initial segmentation masks and an example of how we draw Ilastik training sample sets: The first label (or class) is dedicated to the internal part of the cells, the second label is used to delineate cell borders, and the third is used for everything else. On the right-hand side is an example of the pixel-wise classification output generated by Ilastik after training on our drawn training set input. While the result is not perfect, this approach allowed us to generate a large number of potential training samples with minimal manual effort. (B) The Ilastik output was then processed via Matlab, where a few simple mathematical morphology operations were used to remove small isolated pixel regions that have been misclassified. We then performed a watershedding operation from the first "internal" label/class into the second "border" class of the Ilastik output. This operation segments separated cell regions in the Ilastik output. The user is then prompted with randomly selected segmentation samples as illustrated in this image. If the user considers the

training sample correct, the user can press the "enter" key and the sample is saved to disk as a training sample for the segmentation U-Net. If not, they can press "q" and the sample is saved to disk in a separate folder should the user want to correct it manually. (C) For tracking set generation, the user is presented with a randomly selected sample from the processed Ilastik output, and a cell from the previous timepoint is randomly selected as the "seed" cell to track as in this image. The user can then click on where they think the cell is in either the "current frame" or "segmentation" image in the window, and a second time if they think the cell has divided. The user input is displayed in the "mother" and "daughter" images in the window, and they can press "enter" or "q" to accept or reject at any time. Both interfaces and the Ilastik output post-processing code are made available with the rest of our code. We intentionally kept the code and the interfaces simple to allow easy modification by others.
(TIF)

**S4 Fig. Segmentation errors identified in our evaluation set.** Only 2 segmentation errors out of 3,422 cells were identified when computing the error rate of our trained U-Net against the ground-truth for our evaluation movie. The error on the left illustrates an over-segmentation error, where the bottom cell has been divided by our algorithm when the ground-truth has not. The error on the right illustrates under-segmentation where two cells in the ground-truth have been identified as a single one by DeLTA. For the second error, the DeLTA output could arguably be considered to be correct and the ground-truth erroneous, as the exact point of cell division is sometimes difficult to discern.
(TIF)

**S5 Fig. Tracking errors identified in our evaluation set.** 31 errors were identified out of 3,073 tracking events in the evaluation movie acquired in our lab. Here we show seven representative errors. For each set of images, the raw input images are shown on the left for the current and previous timepoint. The two images in the center illustrate the DeLTA tracking output. Each cell is color coded and the color is preserved from one timepoint to the next. Divisions are identified with black solid lines. Tracking errors are signaled by red arrows. On the right, the ground-truth data are shown for reference. Note that most errors occur as cells are being flushed out of the chambers.
(TIF)

**S6 Fig. Representative examples of yeast segmentation and tracking with DeLTA.** The yeast time-lapse movies datasets are from the Cellstar sample datasets. Yeast cells were imaged with a 60X phase contrast objective in a microfluidic device that constrained them to a monolayer. Black arrows highlight tracking and segmentation errors. Note that the original training sets did not feature mother-daughter relationship information so we did not train the tracking U-Net to identify daughter cells. The two U-Net models were not trained on data from the experiment shown here, but on other experiments in the dataset. While almost all cells are correctly segmented, tracking appears to be more erroneous. However, the datasets that were available online were significantly smaller than the mother machine movies we had (In total there are 170 movie frames in the yeast dataset, with most of them containing only a few cells, when the mother machine movies contain thousands of frames and tens of thousands of cells.) and therefore the training sets may not be diverse enough for DeLTA to be able to generalize tracking rules to new data. We anticipate this would improve with additional training data.
(TIF)

**S7 Fig. Data augmentation operations.** In order to simulate changes in illumination between different experiments or different microscopes, we implemented two data augmentation operations. Two representative examples are given here. The first example of data augmentation

(top row) generates a random monotonically increasing Piecewise Cubic Hermite Interpolating Polynomial (PCHIP) curve that is used to distort the intensity histogram of the input image. The second data augmentation example (bottom row) generates another PCHIP curve that is multiplied against image intensity along the y-axis (along the chamber) to simulate imaging artefacts caused by microfluidic features in the phase contrast light path. After multiplication, the image intensity range is mapped linearly back to the original range.
(TIF)

**S1 Table. Leave-one-out evaluation of the DeLTA algorithm against other mother machine software analysis datasets.**
(DOCX)

## Acknowledgments

We thank Nadia Sampaio for her help in generating the mother machine datasets, Vincent Chang for assistance with the yeast data training set, and Daniel Eaton for help with data augmentation operations and other useful suggestions.

## Author Contributions

**Conceptualization:** Jean-Baptiste Lugagne.

**Data curation:** Jean-Baptiste Lugagne.

**Funding acquisition:** Mary J. Dunlop.

**Methodology:** Jean-Baptiste Lugagne, Haonan Lin.

**Project administration:** Mary J. Dunlop.

**Software:** Jean-Baptiste Lugagne.

**Supervision:** Mary J. Dunlop.

**Visualization:** Jean-Baptiste Lugagne, Mary J. Dunlop.

**Writing – original draft:** Jean-Baptiste Lugagne, Mary J. Dunlop.

**Writing – review & editing:** Jean-Baptiste Lugagne, Haonan Lin, Mary J. Dunlop.

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
