## [Decision Letter · Decision Letter 0]

29 Sep 2019

Dear Dr Dunlop,

Thank you very much for submitting your manuscript 'DeLTA: Automated cell segmentation, tracking, and lineage reconstruction using deep learning' for review by PLOS Computational Biology. Your manuscript has been fully evaluated by the PLOS Computational Biology editorial team and in this case also by independent peer reviewers. The reviewers appreciated the attention to an important problem, but raised some substantial concerns about the manuscript as it currently stands.

Both reviewers found the application of UNET for lineage tracking in realtime to be interesting. However, the reviewers raise significant concerns, including a more thorough description of the method and rigorous comparison of the implementation to existing methods, including freely available tools, to demonstrated improved performance.

While your manuscript cannot be accepted in its present form, we are willing to consider a revised version in which the issues raised by the reviewers have been adequately addressed. We cannot, of course, promise publication at that time.

Sincerely,

Anand R. Asthagiri

Associate Editor

PLOS Computational Biology

Mona Singh

Methods Editor

PLOS Computational Biology

[LINK]

Reviewer's Responses to Questions

**Comments to the Authors:**

Reviewer #1: Here, Lugagne et al present DeLTA, a suite of image annotation and deep learning tools for the segmentation and tracking of cells and their lineages from microscopy images of E. coli "mother machine" microfluidic devices. The authors use a standard application of U-Net for cell-object segmentation, and a novel and straightforward application U-Net for lineage tracking. The authors analyze their lineage results by reporting the relationships between various features and detected events demonstrating the utility of these results for downstream analysis.

Overall the manuscript describes a useful method to perform a much-needed task in many image analysis pipelines, but some crucial methodological information is missing or could be better organized.

Major Points:

Overall the results presented here are difficult to judge due to two major reasons:

1) The authors make claims as to why their method could perform better than others but do not compare their method to others using the same dataset (e.g. CellProfiler, or standard tracking algorithms).

2) It is not clear what constitutes an error. Often the error rate for segmentation tasks is reported by a variety of metrics (e.g. the U-Net paper uses IOU, pixel error, and others). Do segmentation results of "only 9 errors of 6,311 segmented cells" mean that only 9 pixels were misclassified as foreground/background/contour pixels? How lineage errors are determined is similarly unclear. There are many ways to measure errors of mis-constructed lineage trees (false positive/false negative assignment, etc). Expanded statistics in this regard would allow for easier evaluation of the performance of the models presented.

Twice the authors mention strategies that were used to mitigate overfitting, but they do not report how these strategies improve results. Therefore it is difficult to determine whether these strategies help or hinder the results of the model. It would be beneficial to demonstrate that the strategies implemented here are an improvement over the vanilla U-net implementation. Additionally in other ML applications, often a "validation" set is used during training to determine stopping criteria over long training periods.

Yeast is not mentioned until the discussion, and information about these experiments is not present in the methods.

Figures S1, S2, S3, S4 are difficult to understand and would benefit from more description of their contents. Ground-truth images to compare against images would be additionally beneficial in this regard.

Are the processed "ground truth" images available so that others may reproduce and improve upon these results?

Minor points:

The first line in the "Segmentation" and "U-Net architecture and training" sections should refer to figure S5.

Figure S5 should have a reference from the manuscript it was adapted from.

The authors state "Once a large enough training set was generated...", although it is not clear what constitutes "large enough".

What were the optimization algorithm (SGD, ADAM, etc) and learning rates used for training?

What does the "xcov" function in Matlab do?

Reviewer #2: The authors present a solution to the curation bottleneck problem, where annotation is tedious and varies in accuracy. They developed tools to generate training datasets (these tools are shared). The authors use these training datasets with the UNET model architecture to generate segmentation masks (with an update to the boundary loss) and extend the UNET to predict lineage tracking for E. Coli grown in the “mother machine” microfluidic device. Predictions are made in real time, as data is collected.

Major critiques:

This significance of this work is the ability to predict lineage in real-time, with high accuracy, and ultimately without need of human intervention. Another significant contribution is that the authors provide the GUI tools for curation and generation of training datasets.

The novelty of this work is in adapting the UNET to predict tracks for mother/daughter lineages. The extension of the UNET for tracking should be described more thoroughly. The generation of training data is well-described. However, the strategy in image pairing, and more importantly, the way the loss metric is defined, which is the heart of the novelty, deserves more discussion within the text (beyond S5 caption and second paragraph of UNET-architecture and training).

Figure 2 can better show the power/point of feature extraction. It needs more explanation of significance of chosen features (linking perhaps to next figure). As is, the last three plots can be combined somehow to be less redundant, perhaps showing multiple features.

Figure 3C needs more explanation. Why is this interesting biologically?

Since most imaging suffers from variations in illumination and including this issue in data augmentation step has an important contribution to how well the model generalizes, it would be helpful to have more detail describing how these non-Keras augmentations are generated.

Minor critiques:

A large effort in this work is generating the curation dataset. While the idea is to not have to do this step more than once, this will be the large fraction of the effort for others adopting the technique to their own datasets as well. It would be nice to see some graphic/illustration of the curation process in a supplemental figure.

The use of the word ‘unsupervised’ in a machine learning paper, referring to the downstream task of predicting segmentation and tracks, can be confusing when the main modeling uses a supervised technique. One example is here: “….well-suited to high-throughput, unsupervised data analysis.”

It would be helpful for the figures captions to summarize the significance of the result rather than state the axes.

There is no in-text reference to figure several of the figure portions. (ex. 1A, 2E)

Overall the novel contributions are interesting, my feedback would be to drive some of the ideas to their conclusions in the text. Some of the figures, ideas, and tools feel presented because they exist and aren’t explicitly tied to the main contribution of the paper which is currently presented as a fast and accurate technique to alleviate curation

bottleneck for tracking (the segmentation part already exists, although they added a small update).

**Have all data underlying the figures and results presented in the manuscript been provided?**

Reviewer #1: Yes

Reviewer #2: Yes

PLOS authors have the option to publish the peer review history of their article (what does this mean?). If published, this will include your full peer review and any attached files.

Reviewer #1: No

Reviewer #2: No

---

## [Decision Letter · Decision Letter 1]

22 Jan 2020

Dear Dr Dunlop,

We are pleased to inform you that your manuscript 'DeLTA: Automated cell segmentation, tracking, and lineage reconstruction using deep learning' has been provisionally accepted for publication in PLOS Computational Biology.

In the meantime, please log into Editorial Manager at https://www.editorialmanager.com/pcompbiol/, click the "Update My Information" link at the top of the page, and update your user information to ensure an efficient production and billing process.

One of the goals of PLOS is to make science accessible to educators and the public. PLOS staff issue occasional press releases and make early versions of PLOS Computational Biology articles available to science writers and journalists. PLOS staff also collaborate with Communication and Public Information Offices and would be happy to work with the relevant people at your institution or funding agency. If your institution or funding agency is interested in promoting your findings, please ask them to coordinate their releases with PLOS (contact ploscompbiol@plos.org).

Thank you again for supporting Open Access publishing. We look forward to publishing your paper in PLOS Computational Biology.

Sincerely,

Anand R. Asthagiri

Associate Editor

PLOS Computational Biology

Mona Singh

Methods Editor

PLOS Computational Biology

Reviewer's Responses to Questions

Comments to the Authors:

Please note here if the review is uploaded as an attachment.

Reviewer #1: I feel that the authors have sufficiently addressed the points of the reviewers.

Reviewer #2: The authors present a solution to the curation bottleneck problem, where annotation is tedious and varies in accuracy. They developed tools to generate training datasets (these tools are shared). The authors use these training datasets with the UNET model architecture to generate segmentation masks and extend the UNET to predict lineage tracking for E. Coli grown in the “mother machine” microfluidic device. Predictions are made in real time, as data is collected.

The authors made significant updates to the detail and clarity in the text improving the definitions and discussion of inputs, outputs, classes, data augmentation, error/scoring. Further, the illustration of each of these subjects is stronger in the main and supplemental figures and their captions. This manuscript presents an interesting way of tracking by adapting the UNET architecture in an elegant way. The design of the tools is modular and should be compatible with other methods of at multiple stages of their workflow pipeline.

All minor comments were updated in the text.

Have all data underlying the figures and results presented in the manuscript been provided?

Large-scale datasets should be made available via a public repository as described in the 

PLOS Computational Biology

data availability policy, and numerical data that underlies graphs or summary statistics should be provided in spreadsheet form as supporting information.

Reviewer #1: Yes

Reviewer #2: None

PLOS authors have the option to publish the peer review history of their article (what does this mean?). If published, this will include your full peer review and any attached files.

Do you want your identity to be public for this peer review?

 For information about this choice, including consent withdrawal, please see our Privacy Policy.

Reviewer #1: No

Reviewer #2: No

---

## [Editor Report · Acceptance letter]

25 Mar 2020

PCOMPBIOL-D-19-01279R1 

DeLTA: Automated cell segmentation, tracking, and lineage reconstruction using deep learning

Dear Dr Dunlop,

I am pleased to inform you that your manuscript has been formally accepted for publication in PLOS Computational Biology. Your manuscript is now with our production department and you will be notified of the publication date in due course.

With kind regards,

Laura Mallard
